# Thermodynamic Derivation of the Reciprocal Relation of Thermoelectricity

**DOI:** 10.3390/e26030202

**Published:** 2024-02-27

**Authors:** Ti-Wei Xue, Zeng-Yuan Guo

**Affiliations:** Key Laboratory for Thermal Science and Power Engineering of Ministry of Education, Department of Engineering Mechanics, Tsinghua University, Beijing 100084, China; xuetiwei@tsinghua.edu.cn

**Keywords:** thermoelectricity, Kelvin relation, Onsager reciprocal relation, equilibrium thermodynamics

## Abstract

The Kelvin relation, relating the Seebeck coefficient and the Peltier coefficient, is a theoretical basis of thermoelectricity. It was first derived by Kelvin using a quasi-thermodynamic approach. However, Kelvin’s approach was subjected to much criticism due to the rude neglect of irreversible factors. It was only later that a seemingly plausible proof of the Kelvin relation was given using the Onsager reciprocal relation with full consideration of irreversibility. Despite this, a critical issue remains. It is believed that the Seebeck and Peltier effects are thermodynamically reversible, and therefore, the Kelvin relation should also be independent of irreversibility. Kelvin’s quasi-thermodynamic approach, although seemingly irrational, may well have touched on the essence of thermoelectricity. To avoid Kelvin’s dilemma, this study conceives the physical scenarios of equilibrium thermodynamics to explore thermoelectricity. Unlike Kelvin’s quasi-thermodynamic approach, here, a completely reversible thermodynamic approach is used to establish the reciprocal relations of thermoelectricity, on the basis of which the Kelvin relation is once again derived. Moreover, a direct thermodynamic derivation of the Onsager reciprocal relations for fluxes defined as the time derivative of an extensive state variable is given using the method of equilibrium thermodynamics. The present theory can be extended to other coupled phenomena.

## 1. Introduction

Thermoelectric phenomena were discovered in the first half of the 19th century. In 1821, Seebeck [1] first observed that a thermoelectric voltage can be induced by temperature difference. It became known as the Seebeck effect. Shortly thereafter, in 1834, Peltier [2] discovered an inverse effect that, under isothermal conditions, an electrical current can cause a temperature difference at the junction. Since then, it has opened the prelude to the research on the theory and application of thermoelectric conversion.

In 1851, Thomson (known later as Lord Kelvin) [3] predicted the third effect with a theoretical analysis of the relation between both of the above effects. The third effect stated that a conductor carrying current will absorb or release heat when a temperature gradient exists. Thomson further identified that these three effects are not independent from each other, and developed the Kelvin relations (including the first and the second Kelvin relations) that correlate the Seebeck, Peltier, and Thomson coefficients by the quasi-thermodynamic method. The second Kelvin relation is a more basic one, and its expression is
(1)Tα=−π,
where *T* is the temperature, *α* is the Seebeck coefficient, and *π* is the Peltier coefficient.

In 1893, the Kelvin relations were first verified experimentally, and from then on, they have been widely accepted and utilized [4,5]. Although Thomson gave the correct Kelvin relations, his proof was considered problematic. Because in Thomson’s quasi-thermodynamic method, the irreversible factors due to transfer processes in thermoelectric phenomena were ignored, which was not acceptable. As stated by Onsager in his famous paper in 1931 [6], “*Thomson’s relation has not been derived entirely from recognized fundamental principles, nor is it known exactly which general laws of molecular mechanics might be responsible for the success of Thomson’s peculiar hypothesis*”. Under such historical background, Onsager developed a deep underlying relation of coupling between irreversible transfer processes [6,7]. It is known as the Onsager reciprocal relation (ORR), for which he was awarded the 1968 Nobel Prize in chemistry. Onsager first defined the concepts of “flux” and “force” for coupling processes. The flux was required to be the time derivative of an extensive state variable, $J_{i}=dA_{i}/dt$, while the force was the partial derivative of entropy with respect to the corresponding state variable, $X_{i}=\partial S/\partial A_{i}$, where *A* denotes an extensive state variable, *t* is the time, and *S* is the entropy. Then, the constitutive relation for each flux was expressed as $A_{i}=\sum L_{ij}X_{j}$, and the ORR was $L_{ij}=L_{ji}$. With the help of the fluctuation theorem and the assumption of microscopic reversibility, Onsager gave a rigorous proof of the ORR [6,7,8]. By 1948, Callen [9] again derived the second Kelvin relation by means of the ORR. Since the ORR appears to be more compatible with actual physical scenarios, Callen’s derivation is seemingly more convincing than Thomson’s.

Even so, a critical issue remains. Thermoelectric phenomena including the Seebeck effect, the Peltier effect, and the Thomson effect were believed to be thermodynamically reversible [10,11]. Then, the Kelvin relation connecting these thermoelectric effects should also be essentially independent of irreversibility. Why was the derivation based on the ORR taking into account irreversibility justified, while the quasi-thermodynamic approach ignoring irreversibility suffered from being questioned? Kelvin’s argument was on the basis of “reversibility” of the thermoelectric effects [11]. To avoid the interference of irreversibility, this study constructs the physical scenarios that do not contain any irreversible factors to develop the phenomenological theory in thermoelectricity. Unlike Kelvin’s quasi-thermodynamic approach, it is a completely equilibrium thermodynamic approach. The reciprocal relations in equilibrium thermodynamics are established, on the basis of which the Kelvin relation is once again derived. Moreover, the reciprocal relations for the fluxes defined as the time derivatives of extensive state variables are derived directly based on the reciprocal relations in equilibrium thermodynamics. This work might be significant for understanding the physical nature of the reversibility of thermoelectric phenomena.

## 2. Reciprocal Relations in Equilibrium Thermodynamics

Equilibrium thermodynamics deals with a variety of energy conversions, each of which has a corresponding basic thermodynamic equation. Vining [12] pointed out that there is a similarity between thermoelectric processes and thermodynamic processes involving particle exchange. The basic thermodynamic equation for a thermodynamic system with particle exchange is
(2)dU=TdS+μdN,
where *U* is the internal energy, *μ* is the chemical potential, and *N* is the number of particles. By analogy, the basic thermodynamic equation for a thermoelectric system is [9,11,12,13]
(3)$$dU=TdS+\mu_{e}dN_{e},$$
where *μ*_e_ is the electro-chemical potential and *N*_e_ is the number of charged particles. *μ*_e_ is composed of two parts, the chemical portion, *μ*_c_, and the electrical portion, *μ*_elec_:(4)$$\mu_{e}=\mu_{c}+\mu_{\mathrm{elec}},$$
*μ*_c_ is a function of the temperature and the particle concentration. *μ*_elec_ is equal to the product of the charge on a particle, *e*, and the ordinary electrostatic potential, *φ* [9,13]:(5)$$\mu_{\mathrm{elec}}=e\varphi .$$ We can rewrite the basic thermodynamic equation of thermoelectricity, Equation (3), under the “entropy” representation:(6)$$dS=\frac{1}{T}dU-\frac{\mu_{e}}{T}dN_{e}.$$ Thermodynamic equilibrium can be considered to be guaranteed by the maximum entropy principle. According to the state axiom, being equivalent to the Schwarz’s criterion of equality of second order derivatives, there is the following relation:(7)$$-{\left(\frac{\partial \frac{1}{T}}{\partial N_{e}}\right)}_{U}={\left(\frac{\partial \frac{\mu_{e}}{T}}{\partial U}\right)}_{N_{e}}.$$ We can construct a new thermodynamic variable, *Z*, by performing the following Legendre transformation on entropy:(8)$$Z\equiv S-\frac{1}{T}U+\frac{\mu_{e}}{T}N_{e}.$$ Then, according to Equation (6), the differential expression of *Z* is
(9)$$dZ=-Ud\frac{1}{T}+N_{e}d\frac{\mu_{e}}{T}.$$
*Z* is a combination of various state variables, where each state quantity satisfies the property of equality of second-order mixed partial derivatives. Then, it can be demonstrated that the second-order mixed partial derivatives of *Z* are also equal:(10)$${\left(\frac{\partial U}{\partial \frac{\mu_{e}}{T}}\right)}_{\frac{1}{T}}=-{\left(\frac{\partial N_{e}}{\partial \frac{1}{T}}\right)}_{\frac{\mu_{e}}{T}}.$$ There are two pairs of conjugate variables in Equation (6) as well as Equation (9):$$\left\{\frac{1}{T},\;U\right\},\;\left\{\frac{\mu_{e}}{T},\;N_{e}\right\}.$$ To obtain the reciprocal relations of equilibrium thermodynamics, firstly the “flux” and the “force” need to be defined in the context of equilibrium thermodynamics. Given Onsager’s definition of flux as the time derivative of an extensive state variable, here, we replace the time derivative with a differential to obtain a definition of flux in equilibrium thermodynamics. Correspondingly, the force in equilibrium thermodynamics can be defined as the differential of the partial derivative of entropy with respect to extensive state variable. Thus, for the above pairs of conjugate variables, the constitutive relations under the “flux” representation are
(11)$$dU=\Gamma_{11}d\frac{1}{T}+\Gamma_{12}d\frac{\mu_{e}}{T},$$
(12)$$d\left(-N_{e}\right)=\Gamma_{21}d\frac{1}{T}+\Gamma_{22}d\frac{\mu_{e}}{T},$$
and the constitutive relations under the “force” representation are
(13)$$d\frac{1}{T}=R_{11}dU+R_{12}d\left(-N_{e}\right),$$
(14)$$d\frac{\mu_{e}}{T}=R_{21}dU+R_{22}d\left(-N_{e}\right).$$

According to Equation (11), the phenomenological coefficient, Г_12_, is expressed as
(15)$$\Gamma_{12}={\left(\frac{\partial U}{\partial \frac{\mu_{e}}{T}}\right)}_{\frac{1}{T}},$$
and according to Equation (12), the phenomenological coefficient, Г_21_, is expressed as
(16)$$\Gamma_{21}=-{\left(\frac{\partial N_{e}}{\partial \frac{1}{T}}\right)}_{\frac{\mu_{e}}{T}}.$$ It can be seen that Г_12_ is exactly the left term of Equation (10), and Г_21_ is exactly its right term. Therefore, the two are equal:(17)$$\Gamma_{12}=\Gamma_{21}.$$ Similarly, the two phenomenological coefficients, *R*_12_ and *R*_21_, are equal based on Equation (7):(18)$$R_{12}=R_{21}.$$ Equations (17) and (18) can be referred to as the reciprocal relations in equilibrium thermodynamics. They express the state properties of thermodynamic variables along with being a reflection of thermodynamic symmetry.

The above theoretical framework can be extended to other thermodynamic coupling processes such as heat–work conversion and heat–moisture conversion. For example, the basic thermodynamic equation for a heat–work system is
(19)dU=TdS−PdV,
where *P* is the pressure and *V* is the volume. We can rewrite Equation (19) under the “entropy” representation:(20)$$dS=\frac{1}{T}dU+\frac{P}{T}dV.$$
According to the state axiom, the second-order mixed partial derivatives of entropy are equal:(21)$${\left(\frac{\partial \frac{1}{T}}{\partial V}\right)}_{U}={\left(\frac{\partial \frac{P}{T}}{\partial U}\right)}_{V}.$$ As with the establishment of Equation (10), the following relation can also be obtained:(22)$${\left(\frac{\partial U}{\partial \frac{P}{T}}\right)}_{\frac{1}{T}}={\left(\frac{\partial V}{\partial \frac{1}{T}}\right)}_{\frac{P}{T}}.$$ We can further construct the constitutive relations for heat–work systems under the “flux” representation:(23)dU=Γ11'd1T+Γ12'dPT,
(24)dV=Γ21'd1T+Γ22'dPT,
and the constitutive relations under the “force” representation:(25)d1T=R11'dU+R12'dV,
(26)dPT=R21'dU+R22'dV.

According to Equation (22), the phenomenological coefficients, Г_12′_ and Г_21′_, are equal:(27)Γ12'=Γ21'.
and according to Equation (21), the phenomenological coefficients, *R*_12′_ and *R*_21′_, are equal:(28)R12'=R21'. Equations (27) and (28) are the reciprocal relations for heat–work systems in equilibrium thermodynamics. 

## 3. Thermodynamic Derivation of the Kelvin Relation

The Seebeck, Peltier, and Thomson effects are represented with related coefficients as equilibrium properties. As we know, the Seebeck coefficient, *α*, was defined as the ratio between the induced thermoelectric voltage in response to a temperature difference when the electric current is zero, and the Peltier coefficient, *π*, was defined as the ratio between the electric current-induced heat current and the electric current when the temperature is constant. Since the thermoelectric effects are thermodynamically reversible, the expressions for α and *π* can be given in the context of equilibrium thermodynamics [14]. Equilibrium thermodynamics correlates forced tendencies with infinitesimal fluxes (virtually reversible). In equilibrium thermodynamics, the electric flux corresponds to the change of the charge quantity and the heat flux corresponds to the change of the reversible heat. Thus, here, the Seebeck coefficient, α, is defined as the change of electric potential caused by a change of temperature when the charge quantity is constant:(29)$$\alpha ={\left[\frac{d\left(\frac{1}{e}\mu_{e}\right)}{dT}\right]}_{eN_{e}},$$
and the Peltier coefficient, *π*, is defined as the change of reversible heat caused by a change of charge quantity when the temperature is constant:(30)$$\pi ={\left[\frac{\delta Q_{\mathrm{rev}}}{d\left(eN_{e}\right)}\right]}_{T}={\left[\frac{TdS}{d\left(eN_{e}\right)}\right]}_{T},$$
where the subscript rev denotes “reversible”, *e* is the elementary charge, 1/*eμ*_e_ is the electric potential of system and *eN*_e_ is the charge quantity.

According the above definitions, the specific expression for the Seebeck coefficient can be obtained based on Equations (13) and (14): (31)$$\alpha =\frac{1}{eT}\left(\mu_{e}-\frac{R_{21}}{R_{11}}\right),$$
and the specific expression for the Peltier coefficient can be obtained based on Equations (3) and (13):(32)$$\pi =\frac{1}{e}\left(\frac{R_{12}}{R_{11}}-\mu_{e}\right).$$ According to Equations (31) and (32), when the reciprocal relation, *R*_12_ = *R*_21_, holds, then the Kelvin relation, *Tα* = −*π*, holds. Similarly, the Kelvin relation can be obtained using the reciprocal relation, Г_12_ = Г_21_, as well. That is, the Kelvin relation is derived again in the context of equilibrium thermodynamics. This reveals the nature of “reversibility” of the Kelvin relation.

Let us go back to Kelvin’s quasi-thermodynamic argument for deriving the Kelvin relation. Kelvin proposed a thermoelectric circuit composed of two kinds of materials, A and B, as shown in Figure 1. The two junctions between the two materials are in contact with heat sources at different temperatures [15]. Kelvin argued that in a complete cycle of a perfectly reversible kind, the quantities of heat which it takes in at different temperatures are subject to a linear equation, of which the coefficient is the reciprocal of the temperature [16]. Thus, for this thermoelectric circuit that does not take into account any irreversible factors, the contributions to heat from the Peltier effect and from the Thomson effect cancel each other out [17]:(33)$$I\frac{\pi \left(T\right)}{T}-I\frac{\pi \left(T+\Delta T\right)}{T+\Delta T}-I\int_{T}^{T+\Delta T}\frac{\sigma_{B}-\sigma_{A}}{T}dT=0,$$
where *I* is the electric current and *σ* is the Thomson coefficient. Based on Equation (33), together with the energy conservation law, Kelvin derived the well-known Kelvin relation for the first time [18]. In fact, Equation (33) expresses that the net change in entropy for a reversible thermoelectric cycle is zero [19]:(34)$$\oint dS=\oint \frac{\delta Q_{\mathrm{rev}}}{T}=0$$ Therefore, Equation (33) was referred to as the “Clausius equality” in the thermoelectric cycle [18]. The cyclic integral of entropy equal to zero is a reflection of its state property. Although the concept of entropy had not yet been established at that time, Kelvin had already exploited the state property of entropy ahead of time. As mentioned earlier, the Kelvin relation can also be derived by the reciprocal relation, *R*_12_ = *R*_21_, the essence of which is that the second-order mixed partial derivatives of entropy are equal. The latter is likewise a reflection of the state property of entropy. Mathematically, the equality of the second-order mixed partial derivatives of a function and its cyclic integral equal to zero are equivalent. The properties correlated by a state axiom imply reversibility and equivalency of the second-order mixed partial derivatives of a thermodynamic variable. Likewise, the Kelvin relation with the cyclic integral of zero implies state-like functions and reversibility and, in turn, the equality of the second-order mixed partial derivatives. This explains why Kelvin was able to obtain the correct Kelvin relation. In a sense, the Kelvin relation is a reflection of the state property of entropy.

## 4. Thermodynamic Derivation of Onsager Reciprocal Relations for Transient Processes

In Onsager’s phenomenological theory, the flux was defined as the time derivative of an extensive state variable, while the force was the time derivative of entropy with respect to the corresponding state variable. Onsager gave a rigorous proof of the reciprocal relations for this choice of force–flux pairs by means of the fluctuation theorem and the assumption of microscopic reversibility. Onsager’s derivation, although elegant, is complicated, since it involves micro-statistical theory [11]. Callen [9] pointed out that Onsager’s choice of force–flux pairs corresponds to a special type of transient processes that occur immediately after a system in equilibrium is released from a set of constraints. The reciprocal relations for Onsager’s choice might be related to the state property of the system in equilibrium. Therefore, it is possible to use the method of equilibrium thermodynamics to prove the reciprocal relations for the transient processes.

Based on Equation (6), the constitutive relations under Onsager’s definitions of flux and force can be expressed as
(35)$$\frac{dU}{dt}=L_{11}\frac{1}{T}+L_{12}\frac{\mu_{e}}{T},$$
(36)$$-\frac{dN_{e}}{dt}=L_{21}\frac{1}{T}+L_{22}\frac{\mu_{e}}{T}.$$ According to Equation (35), the phenomenological coefficient, *L*_12_, is expressed as
(37)$$L_{12}={\left(\frac{\partial \frac{dU}{dt}}{\partial \frac{\mu_{e}}{T}}\right)}_{\frac{1}{T}},$$
and according to Equation (36), the phenomenological coefficient, *L*_21_, is expressed as
(38)$$L_{21}=-{\left(\frac{\partial \frac{dN_{e}}{dt}}{\partial \frac{1}{T}}\right)}_{\frac{\mu_{e}}{T}}.$$ Changing the order of derivatives gives
(39)$$L_{12}=\frac{d}{dt}{\left(\frac{\partial U}{\partial \frac{\mu_{e}}{T}}\right)}_{\frac{1}{T}},$$
(40)$$L_{21}=-\frac{d}{dt}{\left(\frac{\partial N_{e}}{\partial \frac{1}{T}}\right)}_{\frac{\mu_{e}}{T}}.$$ It can be seen that *L*_12_ is exactly the time derivative of Г_12_, and *L*_21_ is exactly the time derivative of Г_21_:(41)$$L_{12}=\frac{d\Gamma_{12}}{dt},$$
(42)$$L_{21}=\frac{d\Gamma_{21}}{dt}$$

Since Г_12_ = Г_21_ holds, *L*_12_ = *L*_21_ holds. The ORR for the transient processes is guaranteed by the symmetric reciprocal relations in equilibrium thermodynamics. That is, the reciprocal relations developed by Onsager can be directly derived by the approach of equilibrium thermodynamics. This provides a distinctive insight into Onsager’s phenomenological theory. As Goddard [20] claimed, *Onsager’s symmetry is simply a reflection of the underlying symmetry of equilibrium thermodynamics*.

Note that the condition where the fluxes are defined as the time derivatives of extensive variables refers to the transient processes, which is the key to the reciprocal relations being able to be proved [21,22]. However, the above derivation does not hold for the force–flux pairs that refer to the steady-state processes [9,23]. Callen [11] pointed out that the application of the reciprocal relations of transient processes to steady-state processes involves a fundamental approximation. The reciprocal relations of steady-state processes have so far not been rigorously proven and remain controversial. Transient processes represent the tendency “off” the equilibrium (i.e., time gradients at the equilibrium within infinitesimal time) and are governed by (symmetric) reversible processes. However, the steady-state processes further away from equilibrium tendency are irreversible in nature. There might be an essential difference between the reciprocal relations of steady-state and transient processes, which needs to be further explored.

## 5. Conclusions

According to the state axiom, the second-order mixed partial derivatives of a thermodynamic variable are equal. The reciprocal relations of thermoelectricity in equilibrium thermodynamics are established based on this thermodynamic principle. Moreover, the Kelvin relation is derived again successfully using the reciprocal relations in equilibrium thermodynamics. 

In Kelvin’s quasi-thermodynamic argument, the establishment of the Kelvin relation employed the property that the cyclic integral of entropy is zero. Mathematically, the equality of the second-order mixed partial derivatives of a function and its cyclic integral equal to zero are equivalent. This explains why Kelvin was able to obtain the correct Kelvin relation. In a sense, the Kelvin relation is a reflection of the state property of entropy.

The reciprocal relations under the conditions where the fluxes are defined as the time derivatives of extensive variables and the forces are the derivatives of the entropy with respect to these same state variables are derived by means of the reciprocal relations in equilibrium thermodynamics. The reciprocal relations for this choice of force–flux pairs reflect the underlying symmetry of equilibrium thermodynamics.

Thermodynamic derivation applies only to the fluxes defined as time derivatives of extensive variables that refer to a type of transient processes and does not hold for most of the force–flux pairs in steady-state processes. The application of the reciprocal relations of transient processes to steady-state processes might involve new physical aspects pending further exploration.

## Figures and Tables

**Figure 1 entropy-26-00202-f001:**
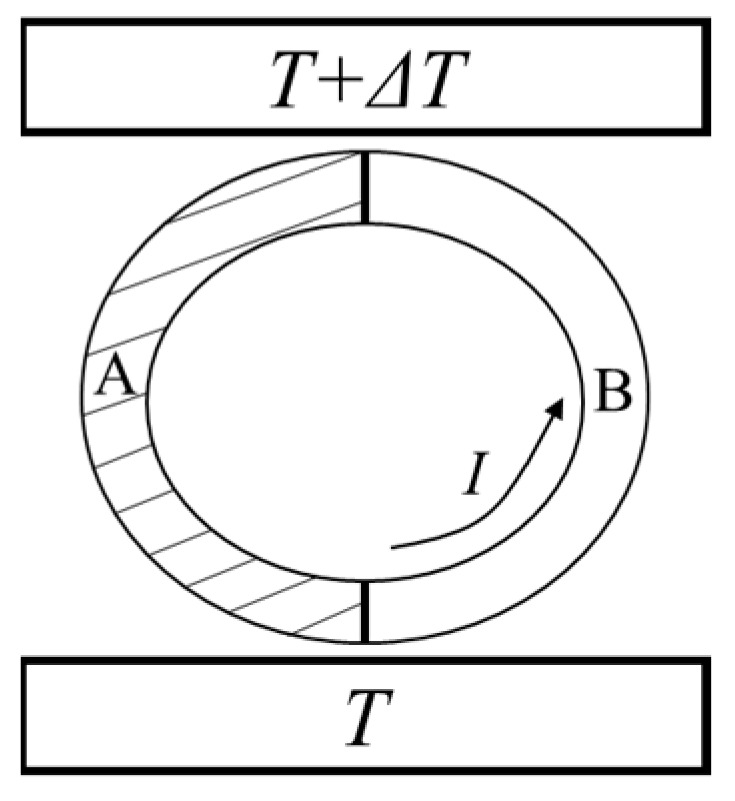
Kelvin’s thermoelectric circuit [15].

## Data Availability

Data are contained within the article.

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
