# Peer review of "Thermodynamic Derivation of the Reciprocal Relation of Thermoelectricity"

_entropy, 2024, doi:10.3390/e26030202_

Round 1

Reviewer 1 Report

Comments and Suggestions for Authors

This manuscript presents creative analyses and alternative derivations of thermoelectric (Seebeck, Peltier, and Thomson) coefficients and the Onsager Reciprocal Relations. Critical analysis and comparison with related references are provided. It will contribute to a better understanding of “the equality of certain ratios between flows and forces in thermodynamic systems out of equilibrium, but where a notion of local equilibrium exists.”

I recommend the manuscript be published.

The following comments are to enlighten the authors' statements and may be used if so desired by the authors to finalize their manuscript:

·       The reversible analysis correlates the properties of initial and final states using hypothetical reversible processes for convenience but its results are valid in general since the state properties are independent on the process path or if irreversible, but only on the final and initial state themselves.

·       Kelvin thermoelectric relation is near-reversible (“independent of irreversibility”), as are the Onsager Reciprocal Relations, ORR (out of equilibrium but with “notion of local equilibrium”).

·       The Seebeck, Peltier, and Thomson effects are represented with related coefficients as [equilibrium] properties.

·       Equilibrium thermodynamics correlates forced-tendency with infinitesimal fluxes (virtually reversible).

·       Transient processes represent the tendency “off" the equilibrium (i.e., time gradients at the equilibrium within infinitesimal time dt) and are governed by [symmetric] reversible processes. However, the steady-state processes further away from equilibrium tendency are fundamentally different and irreversible in nature.

·       The properties correlated by a state axiom imply reversibility and equivalency of the second-order mixed partial derivatives of a thermodynamic variable. Likewise, the Kelvin relation with the cyclic integral of zero implies state-like functions and reversibility, and in turn the equality of the second-order mixed partial derivatives.

Reviewer 2 Report

Comments and Suggestions for Authors

See attached file.

Reviewer 3 Report

Comments and Suggestions for Authors

The reviewed ms. by Xue and Guo "Thermodynamic derivation of the reciprocal relation of thermoelectricity" belongs to a class of the so-called fundamental studies, or more accurately, studies on fundamentals of thermodynamics as viewed by its well-improved phenomenological approach. The Authors describe thermoelectricity as offered in the 19th century by Seebeck, Peltier and Lord Kelvin (W. Thomson), and extended by Lars Onsager in the mid-twenty centuries. 

The Authors, however, follow their path of rationale based on thermodynamic equilibrium and reversibility, Fig. 1. They try to improve - as they express it - a quasi-thermodynamic rationale offered by Kelvin to achieve the same, as Kelvin derived, relation, namely Eq. (1). They wish, however, to develop their line of argumentation as immersed in the tight conceptual connection of the well-celebrated thermodynamic "couple": (thermodynamic) equilibrium and reversibility. 

The paper contains a few inconsistencies. For example, in sec. 1 (Introduction) The Authors claim, on the one hand, that "Because in Thomson’s quasi-thermodynamic method, the irreversible factors due to transfer processes in thermoelectric phenomena were ignored ..." (p. 2, beginning), then, at the end of p. 2 they state: "However, he [Lord Kelvin] placed his argument under the practical scenario that irreversible factors could not be ignored ...", thus, the present reviewer remains in doubt which was actually the opinion/attitude of Lord Kelvin? 

The real challenge of the reviewed study appears to be resting (by the Authors) on the equilibrium, only. To some extent, resting on it, and consequently performing the evaluation at equilibrium, can be accepted upon realizing that typically (more: molecularly) the thermodynamic equilibrium is realizable using reversibility viewed, however, more in the statistical sense - the pathway that the Authors wished likely to avoid. Their derivation, based on the 'conceptual couple' equilibrium-reversibility, looks legitimate, albeit it violates somehow Onsager's (and, others) fundamental assumptions that the flux-force thermodynamic relations belong to linear thermodynamics of nonequilibrium processes, realizable close to thermodynamic (local) equilibrium. (Another doubt would be to use time derivatives, see Eqs. (39)-(42) whixh is not a customary procedure while at equilibrium.) On the contrary, the Authors propose here their way in which, what is not underscored explicitly, the thermodynamic equilibrium goes via the maximum entropy principle, somehow hidden in their study, see Eqs. (8) and (9).  

When having certain phrases preceded Eqs. (8) and (9), right at "According to the state axiom ..." one then speaks about the 2nd order mixed derivative; for a layman, because at Eq. (7) a first order partial derivative is seen, one could propose to write down "According to the state axiom, being equivalent to the Schwarz's criterion of equality of second order derivatives ..." or alike. 

Technically, there is a flaw at p. 7, see "abstruse micro-statistical theory." which would likely refer to the statistical-thermodynamical theory of ensembles by Gibbs. 

To conclude, the ms. looks like it was a quite independent proposal of deriving the connections between fundamental thermodynamical relations for (un)conjugated fluxes (thermo-electricity) in which, though not microscopically, justified, but the phenomenologically derivable output can attract readers' interest, after slightly improving the presentation. 

Comments on the Quality of English Language

It seems to me (I'm not a native speaker, however) that the quality of English Language is sufficiently correct; at least one exception has been indicated as a technical flaw. 

Round 2

Reviewer 1 Report

Comments and Suggestions for Authors

The authors have updated the manuscript including the suggested comments and it is recommended to be published in present form. As already stated in the prior report, the manuscript will contribute to a better understanding of “the equality of certain ratios between flows and forces in thermodynamic systems out of equilibrium, but where a notion of local equilibrium exists.”

Reviewer 2 Report

Comments and Suggestions for Authors

See attached file.

Reviewer 3 Report

Comments and Suggestions for Authors

Please do improve for aesthetic reasons the 1st sentencje of Abstract.